# How is *Etuaptmumk/*Two-Eyed Seeing characterized in Indigenous health research? A scoping review

**Sophie I. G. Roher** [1,2‡]*, **Ziwa Yu**[3‡], **Debbie H. Martin**[4‡], **Anita C. Benoit**[5,6,7‡]

**1** Social and Behavioural Health Sciences Division, Dalla Lana School of Public Health, University of Toronto, Toronto, Ontario, Canada, **2** Institute for Circumpolar Health Research, Yellowknife, Northwest Territories, Canada, **3** Aligning Health Needs and Evidence for Transformative Change (AH-NET-C): A JBI Centre of Excellence, School of Nursing, Dalhousie University, Halifax, Nova Scotia, Canada, **4** Health Promotion Division, Faculty of Health, Dalhousie University, Halifax, Nova Scotia, Canada, **5** Department of Health and Society, University of Toronto Scarborough, Toronto, Ontario, Canada, **6** Dalla Lana School of Public Health, University of Toronto, Toronto, Ontario, Canada, **7** Women's College Research Institute-Women's College Hospital, University of Toronto, Toronto, Ontario, Canada

‡ SIGR and ZY contributed equally to this work as first co-authors. DHM and ACB also contributed equally to this work as Joint Senior Authors.
* sophie.roher@mail.utoronto.ca

**Data Availability Statement:** All relevant data are within the manuscript and its Supporting information files.

## Abstract

Our scoping review sought to consider how *Etuaptmumk* or Two-Eyed Seeing is described in Indigenous health research and to compare descriptions of Two-Eyed Seeing between original authors (Elders Albert and Murdena Marshall, and Dr. Cheryl Bartlett) and new authors. Using the JBI scoping review methodology and qualitative thematic coding, we identified seven categories describing the meaning of Two-Eyed Seeing from 80 articles: guide for life, responsibility for the greater good and future generations, co-learning journey, multiple or diverse perspectives, spirit, decolonization and self-determination, and humans being part of ecosystems. We discuss inconsistencies between the original and new authors, important observations across the thematic categories, and our reflections from the review process. We intend to contribute to a wider dialogue about how Two-Eyed Seeing is understood in Indigenous health research and to encourage thoughtful and rich descriptions of the guiding principle.

## Introduction

*Etuaptmumk*, the Mi'kmaq word for "the gift of multiple perspectives" is often called by the English term 'Two-Eyed Seeing'. *Etuaptmumk*, or Two-Eyed Seeing, is a guiding principle that was introduced to the academic community by M'ikmaq Elders Albert and Murdena Marshall and Dr. Cheryl Bartlett in Unama'ki (Cape Breton), Nova Scotia [1]. Unlike many theoretical traditions used in academia, which are commonly rooted in European and Western values and philosophies, Two-Eyed Seeing developed from rich Mi'kmaq traditions. Although grounded in Mi'kmaq language and culture, Two-Eyed Seeing reflects overarching concepts that exist

**Funding:** DM is supported by a Tier II Canada Research Chair in Indigenous Peoples' Health and Well-Being through the Canada Research Chairs Program (grant # CRC-2016-00076) (https://www.chairs-chaires.gc.ca/home-accueil-eng.aspx); AB is supported by an OHTN CIHR New Investigator Award through the Canadian Institutes of Health Research (grant # APP268899) (https://cihr-irsc.gc.ca/e/44181.html); and SR was supported by a Doctoral Research Award through the Canadian Institutes of Health Researcher's Institute of Aboriginal People's Health (grant # CIHR/383832) (https://cihr-irsc.gc.ca/e/8668.html). The funders had no role in study design, data collection and analysis, decision to publish, or preparation of the manuscript.

**Competing interests:** The authors have declared no competing interests exist.

within diverse Indigenous communities, whose knowledge systems, although distinct and linked intimately to the unique geographic territories from which they have emerged, share ontological insights [2, 3] (S1 File). In 2004, Two-Eyed Seeing gained recognition in academia when Elders Albert and Murdena Marshall introduced the principle to guide an Integrative Science research project that was part of Dr. Bartlett's Canada Research Chair in Integrative Science. The research fed back to an academic program in Integrative Science, a science degree program, which seeks to bring together Indigenous and Western knowledges and ways of knowing [4–6]. Since then, Two-Eyed Seeing usage has extended well beyond the realm of Integrative Science (which is based largely within the disciplines of biology, ecology, and environmental science), including its frequency of use within Indigenous health scholarship. With increase in usage, Two-Eyed Seeing has been described in inconsistent and sometimes contradictory ways [7–9]. Our review seeks to explore how Two-Eyed Seeing is being depicted in Indigenous health research to encourage rich descriptions of the guiding principle.

## History of Two-Eyed Seeing

The late Chief Charles Labrador, a former leader of Acadia First Nation, used the metaphor of diverse species of trees living together in a forest; despite their differences, their roots are connected underground, and are "holding hands". He used this metaphor as a guide for how to be in the world [2, 3]. Going forward from this teaching, Mi'kmaq Elders Albert and Murdena Marshall from Eskasoni First Nation introduced the term *Two-Eyed Seeing* to describe the process of bringing together the strengths of Indigenous and Western knowledges for the benefit of all [3, 10]. Elders Albert and Murdena Marshall worked closely with Dr. Cheryl Bartlett, a non-Indigenous professor of biology who held a Canada Research Chair in Integrative Science, to bring Two-Eyed Seeing into Dr. Bartlett's research program and, consequently, into an Integrative Science academic program [2, 5, 6]. The term *Integrative* was chosen intentionally—it is verb or action-based, which reflects the active, ongoing nature of Indigenous knowledge systems; they are forever changing, developing, and transforming in response to what is happening in the world. Since that time, Two-Eyed Seeing has gained momentum in the field of health. This is partly due to the fact that in 2014, Two-Eyed Seeing was included in the five-year strategic plan released by the Canadian Institutes of Health Research's (CIHR) Institute for Aboriginal People's Health, informing funding opportunities and the overall vision for the future of Indigenous health [11, 12]. In 2017, it also featured in the Naylor report, the Canadian government's fundamental science review [13].

## Rationale

Two-Eyed Seeing has been increasingly employed in published Indigenous health scholarship. It is interesting to note that scholars tend to interpret Two-Eyed Seeing in various and sometimes contradictory ways [7, 8, 14]; some suggest that Two-Eyed Seeing encourages the *bridging* of Indigenous and Western knowledges, while others see Two-Eyed Seeing as encouraging the *blending* or *integration* of Indigenous and Western knowledges [7, 15–19]. An array of terms have been used to characterize Two-Eyed Seeing, including *theoretical framework*, *concept*, *methodology*, and *guiding principle*. Each of these terms carries very different meanings and has consequences for how the guiding principle is used in research studies. For example, a methodology is largely focused on technique, whereas a theoretical framework may provide the logic behind methodological choices. While both provide insight into complex and competing worldviews [20], characterizing Two-Eyed Seeing as a methodology versus a theoretical framework can imply very different usage based on their definitions.

There have been two recent reviews that look at the way that Two-Eyed Seeing has been used in research articles. Wright et al. [7]'s integrative review examines how Two-Eyed Seeing has been interpreted and operationalized by researchers (n = 37) and Forbes et al. [14]'s scoping review considers how researchers have applied the concept of Two-Eyed Seeing in primary studies related to Indigenous health (n = 22). Both authors highlight inconsistencies in how Two-Eyed Seeing is taken up by researchers; however, the main focus of these reviews is on the *use* and *application* of Two-Eyed Seeing. Rather than identify how Two-Eyed Seeing is being applied and adopted in health research, our study seeks to describe and compare the language used by the original and new authors to represent Two-Eyed Seeing. We believe that this language provides insight into how authors understand the guiding principle, and it suggests how their understandings change over time.

When new concepts or frameworks are introduced into academia and taken on in various research projects, it is not unusual that they would morph and transform with time and context. For example, intersectionality is a concept emerging from ideas debated in critical race theory [21]. It has since then been adapted to meet diverse disciplinary needs, leading many scholars to engage with it theoretically and methodologically [22]. Similarly, feminist ideologies and movements have evolved to include several waves recognizing white feminist theories and later multiracial feminist theories [23, 24]. This morphing and transformation is also often observed when prescriptive guidelines for their use have not been developed or implemented. Similar to research approaches such as community-based participatory research [25–27], there is no 'one size fits all' approach to using Two-Eyed Seeing; it is taken up by different authors in a variety of forms. The flexibility of Two-Eyed Seeing is one of its strengths; it both allows the guiding principle to contribute to various projects in novel and tailored ways and provides the opportunity for it to evolve and advance over. time. Nevertheless, when Two-Eyed Seeing is not described thoroughly or with intention, there is a risk that it could be misrepresented or perhaps diminished. Indeed, Elder Albert Marshall emphasized this concern in his article *Learning Together by Learning to Listen to Each Other* [28]. He underscored that the work of Two-Eyed Seeing is not easy:

> "The work can all too easily slip into a lazy, tokenistic approach in which *Etuaptmumk*/ Two-Eyed Seeing and similar efforts quickly become mere jargon, trivialized, romanticized, co-opted, or used as a "mechanism" where pieces of knowledge are merely assembled in a way that lacks the S/spirit of co-learning"

[28].

In order to mitigate against this concern, it is important to better understand how Two-Eyed Seeing has been described in Indigenous health research literature and whether these descriptions are consistent with the ways that Elders Albert and Murdena Marshall and Dr. Cheryl Bartlett characterized Two-Eyed Seeing in their own presentations and writing. The purpose of our review is (1) to describe how Two-Eyed Seeing is defined in Indigenous health research and (2) to compare these representations with the original authors' descriptions of Two-Eyed Seeing.

Our review does not intend to criticize researchers who described Two-Eyed Seeing in a way that is different from our own understandings. Rather, we hope to contribute to a wider dialogue about the way that Two-Eyed Seeing is taken up in health research and to encourage thoughtful and in-depth descriptions of the guiding principle so that it remains robust and strong with time.

## Methods

Our scoping review used the methods outlined in the JBI Manual for Evidence Synthesis [29]. The research question was: What are the descriptions of *Etuaptmumk*/Two-Eyed Seeing in Indigenous health research? We also used the Preferred Reporting Items for Systematic Reviews and Meta-analyses extension for Scoping Reviews (PRISMA-ScR) [30] to guide the conduct and reporting of this scoping review.

### Data sources and search strategy

A JBI-trained health science librarian assisted in the development and implementation of search strategies in six bibliographic databases: PubMed (MEDLINE), Academic Search Premier (EBSCOhost), PsycINFO (EBSCOhost), CINAHL (EBSCOhost), Bibliography of Native North American (EBSCOhost), and EMBASE. Searches for grey literature were conducted in ProQuest Dissertations and Theses, Indigenous Studies Portal (iPortal) and the Institute for Integrative Science & Health (IISH) website. Publications in four key journals (International Journal of Indigenous Health, International Journal of Circumpolar Health, International Indigenous Policy Journal, and Pimatisiwin: A Journal of Aboriginal and Indigenous Community Health) were screened for eligible studies. Specifically, we used the search function on journal websites by entering keywords 'Two-Eyed Seeing' and '*Etuaptmumk*' and screening the search results by titles and abstracts for relevancy. Reference lists of included literature were scanned for additional records. Finally, we contacted authors of conference abstracts to request potential full-text publications. See S1 Table for full search strategies.

### Eligibility criteria

The key concept underpinning our scoping review is 'Two-Eyed Seeing' known in the Mi'kmaq language as *Etuaptmumk*. The populations of interest were First Nations, Inuit and Métis in Canada, though we also considered international literature that defined Two-Eyed Seeing. We included literature with non-Indigenous populations (e.g., non-Indigenous health professionals, educators, policy makers, and researchers) so long as the purpose of research was related to our definition of Indigenous health research.

According to the CIHR, Indigenous health research can be defined by "any field or discipline related to health and/or wellness that is conducted by, grounded in, or engaged with, First Nations, Inuit or Métis communities, societies or individuals and their wisdom, cultures, experiences or knowledge systems, as expressed in their dynamic forms, past and present" [31]. In keeping with the CIHR definition, we considered literature across a variety of academic disciplines including health sciences, education, biology, among others.

Only English-language literature published since the year 2004 was included, as this is the year that Two-Eyed Seeing was first introduced to the Integrative Science research program by Elders Albert and Murdena Marshall [32]. Eligible literature included: primary research of all study designs, all types of reviews, text and opinion papers, theses and dissertations, newsletters, organizational and governmental reports, and conference proceedings. For conference proceedings, only abstracts that have a corresponding full-text publication (e.g., journal articles, unpublished manuscripts, reports, etc.) were included.

### Citation management and screening process

Following the search, all identified citations were first imported to EndNote and the majority of duplicates were removed. The remaining citations were then uploaded into Covidence (Veritas, Melbourne, Australia), an online systematic review platform, which removed

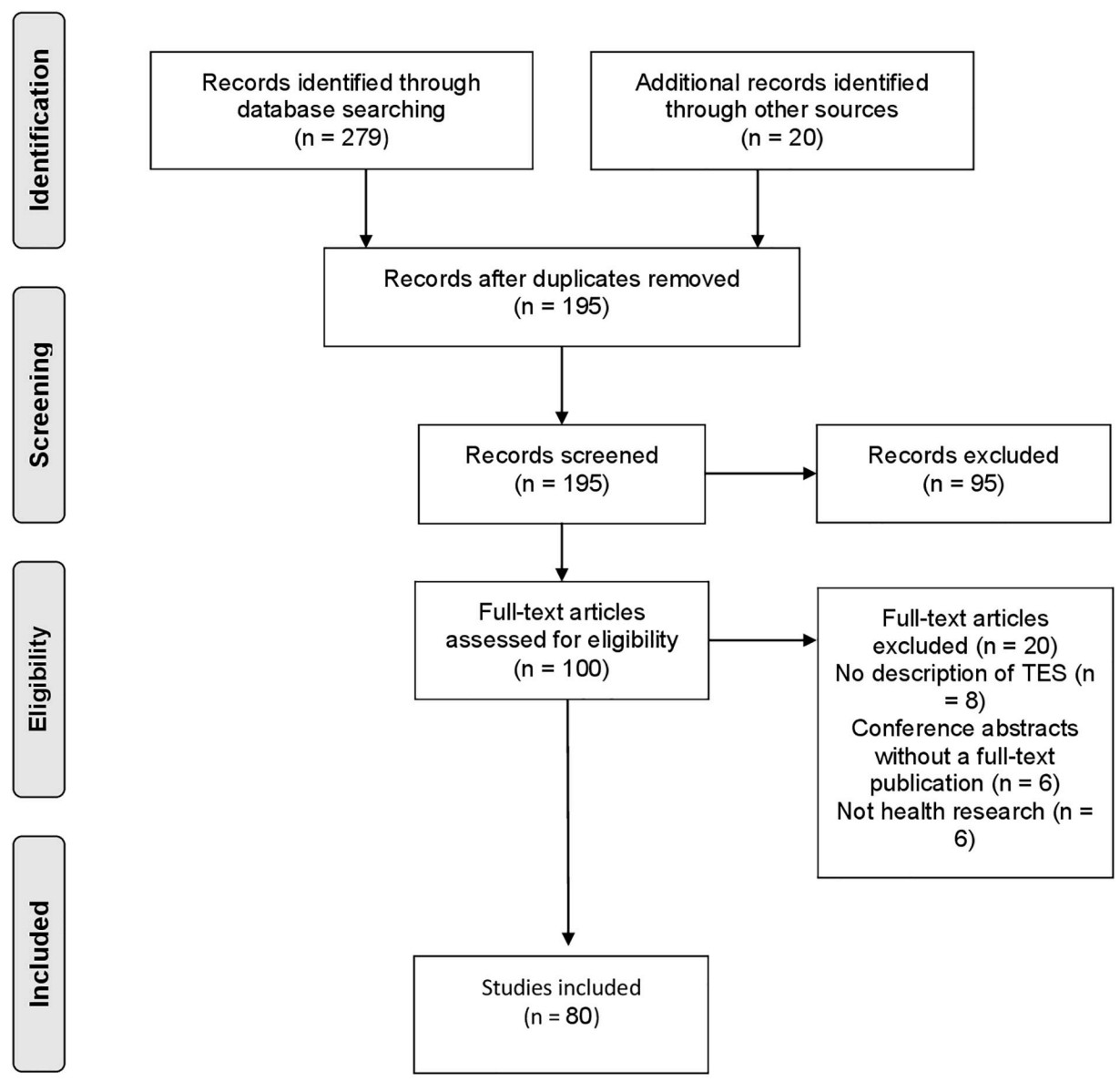

**Fig 1. PRISMA-ScR flow diagram.** Study selection process.

remaining duplicates across databases. A team of four reviewers worked collaboratively on Covidence to screen titles and abstracts independently against pre-defined eligibility criteria. Potentially relevant citations were then retrieved in full-text and uploaded into Covidence. Two reviewers independently assessed the eligibility of full-text literature and provided a rationale for exclusion. Reasons for exclusion are summarized in Fig 1. Throughout the screening process, disagreements between reviewers were settled through discussion or by a third reviewer, if required.

### Data extraction and synthesis

We extracted the data in two parts. First, a charting table in Microsoft Excel was created by authors to extract the following details from included literature: author(s), year of publication,

literature type, research design, aim of research, study participants, geographical location, and whether the term *Etuaptmumk* was used. Before formal data extraction, the charting table was trialed independently by each member of the review team on four full-text articles to ensure all relevant information was captured and that there was a mutual understanding of all data extraction terms. The remaining data extraction was completed with two members of the review team independently reviewing each article. Any disagreements were resolved through discussion or with a third reviewer, if required.

Second, we extracted quotations that describe how authors were interpreting the meaning of *Etuaptmumk*/Two-Eyed Seeing in health research. The quotations were pooled and analyzed using qualitative thematic coding [33]. To start, two reviewers independently extracted a verbatim quotation from each paper. All verbatim quotations were then read multiple times and grouped into categories according to conceptual similarity. The entire review team met multiple times to discuss the name and meaning of these themes until consensus was reached. In the end, a brief description was created for each theme to represent the inclusive meaning of the quotations that are captured within that theme. Since the goal of scoping reviews is to map existing evidence regardless of quality, we did not assign levels of credibility to extracted findings. All data extraction steps were completed in Microsoft Excel.

## Results

The literature screening process is summarized in a PRISMA-ScR diagram (Fig 1). The search of databases resulted in 279 records. An additional 20 articles were included based on searching key websites, journals, and reviewing relevant articles' reference lists. After removing duplicates, 195 remained for title and abstract screening, of which 100 progressed to full text review. Ultimately, 80 articles met the eligibility criteria and were included in our review.

### Characteristics of included articles

An overall description of included literature is presented in S2 and S3 Tables. Of the 80 articles included, nine were authored by Albert Marshall, Murdena Marshall, and Cheryl Bartlett (individually or as part of a team) [2–5, 10, 28, 34–36]. The remaining 71 articles were written by other authors [8, 9, 15–19, 37–100]. All 80 articles were published after 2007, 65 of which were published in the last five years (i.e., 2015 onward). The majority of included research were journal articles (n = 61), followed by theses and dissertations (n = 9), magazine articles (n = 4), book chapters (n = 4), reports (n = 1) and unpublished manuscripts (n = 1). In terms of research design, 43 research articles employed an empirical design; among these, 14 used community-based research design variations, such as community-based participatory research or participatory action research [8, 9, 19, 39, 45, 49, 52, 69, 84–86, 91, 92, 101]. The 37 non-empirical articles were discussion papers, literature and systematic reviews, policy analyses, and interview transcription. Most empirical studies involved First Nations participants, while a few explicitly mentioned Inuit and/or Métis communities [18, 39, 42, 50, 51, 60, 61, 72, 74, 75, 84].

The geographical locations of literature were classified according to the study sites of empirical research, the home bases of the specific programs/strategies of interest, and the relevant regions for policies and reviews (S2 and S3 Tables). If none of the above criteria applied, the country of the first author was recorded. Most research was based in Canada (n = 71). Among these, 16 were in Ontario (n = 16) and 14 were in Nova Scotia.

We extracted whether the term 'Etuaptmumk' was included in the eligible articles. 'Etuaptmumk' is more closely translated in English to the 'gift of multiple perspectives'. This is important in fully understanding the Two-Eyed Seeing guiding principle, which expands it beyond

Indigenous and non-Indigenous perspectives. The 'real' translation 'gift of multiple perspectives' requires far more explanation and teaching before it could be understood in English. The term 'Etuaptmumk' also establishes the connection to the Mi'kmaq worldview.

## Descriptive categories of *Etuaptmumk* /Two-Eyed Seeing

From the 80 articles, seven categories were identified that describe the meaning of *Etuaptmumk*/Two-Eyed Seeing: guide for life, responsibility for the greater good and future generations, co-learning journey, multiple or diverse perspectives, spirit, decolonization and self-determination, and humans being part of ecosystems. S4 and S5 Tables provide a summary of extracted findings created from combining the original and new authors descriptions of Two-Eyed Seeing to support each of the seven categories. Below we outline in greater detail each of the seven categories and provide a quotation from one of the papers reviewed to help illuminate the meaning of the category.

**Category 1: Guide for life.** Two-Eyed Seeing is a way of living that transcends disciplinary and academic boundaries to guide all aspects of life. One can draw upon Shawn Wilson's descriptions of the difference between ontology, epistemology, methodology, and axiology to help describe Two-Eyed Seeing's relationship with these terms. Ontology is the theory of reality, epistemology is the theory of how we come to know something, methodology is the theory of how knowledge is obtained, and axiology is the ethics or moral that guide knowledge gathering [102]. Two-Eyed Seeing is not solely an ontology, epistemology, methodology, or axiology. It exists at all levels, it is more wholistic; it is a way of knowing, being, doing, and seeing [1]. Two-Eyed Seeing, as a guiding principle, is mental, spiritual, physical, and emotional. When applied to the context of diverse knowledges, Two-Eyed Seeing entails an ongoing reflection on self, openness to new perspectives, and readiness for adjustment. As Elder Albert Marshall describes: "Two-Eyed Seeing is hard to convey to academics as it does not fit into any particular subject area or discipline. Rather, it is about life: what you do, what kind of responsibilities you have, how you should live while on Earth. . . i.e., a guiding principle that covers all aspects of our lives: social, economic, environmental, etc." [5].

**Category 2: Responsibility for the greater good and future generations.** Two-Eyed Seeing entails a sense of responsibility for the benefit of all. It calls upon all people, Indigenous and non-Indigenous alike, to reposition the work we are doing to think about its value and impact seven generations ahead. Two-Eyed Seeing motivates people to use all knowledges available and take action in making the world a better and healthier place. Bartlett et al. offer a widely cited description of this point: "Two-Eyed Seeing intentionally and respectfully brings together our different knowledges and ways of knowing, to motivate people, Aboriginal and non-Aboriginal alike, to use all our gifts so we leave the world a better place and not comprise the opportunities for our youth (in the sense of Seven Generations) through our own inaction" [103].

**Category 3: Co-learning process.** Two-Eyed Seeing is a continual process of co-learning, relationship-building, and adaption. It encourages ontologically different people to engage in conversation and put their own knowledges and actions forward for examination. Elder Albert Marshall has pointed out that, "we need to embark on a co-learning journey of Two-Eyed Seeing in which our two paradigms (western and Indigenous) will be put on the table to be scrutinized. We need to honestly be able to say that the essence, the spirit of our two ways, has been respected as we work to balance the energies of those ways" [as quoted in 80]. Depending on the people involved, the specific steps of co-learning will differ and there might not always be consensus. To embody the spirit of co-learning, one must show a commitment to the process of conversation rather than focus only on the outcomes. As Hall et al. assert: "It must be kept

in mind in any movement forward that two-eyed seeing as a guiding concept is a learning process and not a perfected outcome. It is not a prescriptive list about how to achieve cultural renewal through an Indigenous governed research process" [58].

**Category 4: Multiple or diverse perspectives.** Two-Eyed Seeing offers strategies to understand multiple or diverse knowledges, and to use them constructively side-by-side.

*Respect for multiple realities.* Two-Eyed Seeing advocates for the recognition, acceptance, and respect for diverse perspectives. In the context of Canada's colonial history and oppression of Indigenous cultures, Two-Eyed Seeing articulates a support for Indigenous cultural renewal by creating space for equitable power relationships, where Indigenous knowledges are recognized as valid and useful in their own rights. Two-Eyed Seeing views differences as a gift that bring in new information, rather than a source of clash and tension. As Martin describes: "Two-eyed seeing holds that there are diverse understandings of the world and that by acknowledging and respecting a diversity of perspectives (without perpetuating the dominance of one over another) we can build an understanding of health that lends itself to dealing with some of the most pressing health issues facing Indigenous peoples and communities" [76].

*Perspectives are not static.* Two-Eyed Seeing reminds us of the changing nature of knowledge systems and our own perspectives. This notion is particularly useful in supporting the stance that Indigenous knowledges are dynamic and have wide applicability in the present day. Elders Murdena and Albert Marshall have instructed that "Traditional Knowledge was never meant to stay static and stay in the past. Rather, we must bring it into the present so that everything becomes meaningful in our lives and in our communities" [34]. To solve complex problems, one must caution against settling for one way of doing things but always try to find a better solution. As Elders Murdena and Albert Marshall describe: "The advantage of Two-Eyed Seeing is that you are always fine tuning your mind into different places at once, you are always looking for another perspective and better way of doing things" [5, 34].

*Wholeness/partiality of knowledge.* Two-Eyed Seeing opens up discussion about the completeness of knowledges. In the sense that all knowledges must be respected in their entirety, Two-Eyed Seeing views each knowledge system as distinct and whole, but that each can only ever offer a partial understanding of the world. Donna Martin articulates this point: "Two-eyed seeing does not subsume one way of knowing over another. It encourages fluidity, multiple perspectives, and the use of self-reflection to pose questions and critically consider the partiality of one's perspective" [77]. Thus, when we bring more than one knowledge system together with another, we are increasing what we can know about the world, which is still limited by our experiences as humans. Two-Eyed Seeing encourages us to reflect upon and engage with different perspectives to create a new vision. As Debbie Martin describes: "When both eyes are used together, this does not mean that our view is now 'complete and whole', but that a new way of seeing the world is created that respects the differences that each can offer" [75].

*Co-existence of knowledges.* Two-Eyed Seeing strategically brings knowledges together by taking the strengths of diverse systems in establishing a common ground. It guides people to consciously choose the most suitable knowledge to act upon, depending on the circumstances. This dynamic approach is often referred to as "(inter)weaving" [45, 56, 71, 94], or a "dance" between knowledges [81, 104]. A variety of verbs were used in the included literature to convey the sense of "bringing together", such as blending [15–19], merging [57], integrating [51, 65, 95], and combining [56, 63]. Despite different word use, there is clear articulation in the original authors' publications that knowledges should not be tweaked or merged with one another (i.e., to avoid one knowledge system being subsumed within another) [2, 34].

**Category 5: Spirit.** Two-Eyed Seeing teaches us about the universality of spirit (i.e., there's a spirit in everything) in the Mi'kmaq worldview and thus the importance of including spiritual knowledge in human understandings of the world. Two-Eyed Seeing understands spirit as

essential for a complete person, and that knowledge is gained from the interaction of body, mind, soul and spirit with all aspects of nature. A widely cited quote of Elder Albert stresses the importance of spirit in the Mi'kmaq worldview:

"When you force people to abandon their ways of knowing, their ways of seeing the world, you literally destroy their spirit and once that spirit is destroyed it is very, very difficult to embrace anything—academically or through sports or through arts or through anything—because that person is never complete. But to create a complete picture of a person, their spirit, their physical being, their emotions and their intellectual being. . . all have to be intact and work in a very harmonious way"

[5].

**Category 6: Decolonization and self-determination.** Two-Eyed Seeing makes valuable contributions to decolonizing work by honouring Indigenous perspectives in the processes of how knowledges are created, gathered, and used. For example, Hall et al. (2015) conceptualize Two-Eyed Seeing as the Two Row Wampum,

In my mind I imagine the Indigenous canoe on the river and I imagine the Western canoe. I imagine in the Two- Eyed Seeing process, we're coming together from distinct world views but at the same time within the West, there is that need and that recognized need to decolonize across the disciplines, to challenge as has been done for decades and decades the notions of objectivity and to bring our own subjectivities and stories into our understanding of how knowledge is constructed

[58].

Two-Eyed Seeing also has the potential to advance Indigenous self-determination by nurturing Indigenous leadership in research. Marsh et al. [74]. assert that "[t]he two-eyed seeing approach is consistent with Aboriginal governance, research as ceremony, and self-determination. In other words, it is consistent with the principles of ownership, control, access, and possession."

**Category 7: Humans being part of ecosystems.** Two-Eyed Seeing reinforces a sense of relationality and interconnectedness at two levels: (1) The relationships between diverse groups of people, where sharing cultural differences links people by allowing them to contribute their distinct knowledges to a shared topic; (2) The relationships between humans and nature, where the balance and integrity of larger ecosystems has a direct impact on human health and well-being. The following quote by Heath-Engel (2016) illustrates both of these levels:

The way in which the Two Eyes must work together, is similar to the ways in which all parts of ecosystems, as part of the larger global ecosystem, must work together in harmony, constantly adapting to changes in the overall system, in order for each ecosystem, and the larger global ecosystem to remain healthy. In a similar way, all knowledges about understanding the world must be recognized and appreciated in their own right and entirety, in searching for solutions to the health problems of humanity, in order to for these solutions to be found. This is the core principle of the Two-Eyed Seeing Framework

[62].

## Terminologies

Fig 2 compares the terminologies used by original authors and new authors over time to describe *Etuaptmumk*/Two-Eyed Seeing. Among the original authors, terminologies remained fairly consistent since 2007, with the most frequent being *guiding principle* and *approach*. While the original authors describe Two-Eyed Seeing most often as a *guiding principle* and *approach*, one article by the original authors titled "Two-Eyed Seeing and the Language of Healing in Community-Based Research" also used the terms *Indigenist pedagogy* and *research practice* in reference to Two-Eyed Seeing [2]. We believe that this article with Dr. Marilyn Iwama as the senior author and co-authors Elders Albert and Murdena Marshall, and Dr. Cheryl Bartlett, is one of the primary articles to bridge the original authors' descriptions of Two-Eyed Seeing with health research explicitly. It is one of the first articles to discuss 'health research' directly and to use language that is more commonly used in academic spaces. The language used in this article is notably different from the original authors' other papers and many of the terms are not used by the original authors again. In comparison, the new authors' terminology was highly heterogenous with unique terms emerging each year. The most commonly used terms by new authors included *approach* (used by original authors as well), *framework*, and *concept* (neither of which were used by original authors). *Guiding principle* was also used by new authors, however it was not mentioned in included literature until 2014 (i.e., seven years after Two-Eyed Seeing was introduced to the academic community by the original authors).

## Discussion

This is the first scoping review to characterize and compare descriptions of Two-Eyed Seeing. The plethora of terms used demonstrates a lack of clear consensus amongst authors about how to describe the Two-Eyed Seeing guiding principle. Despite different representations, we

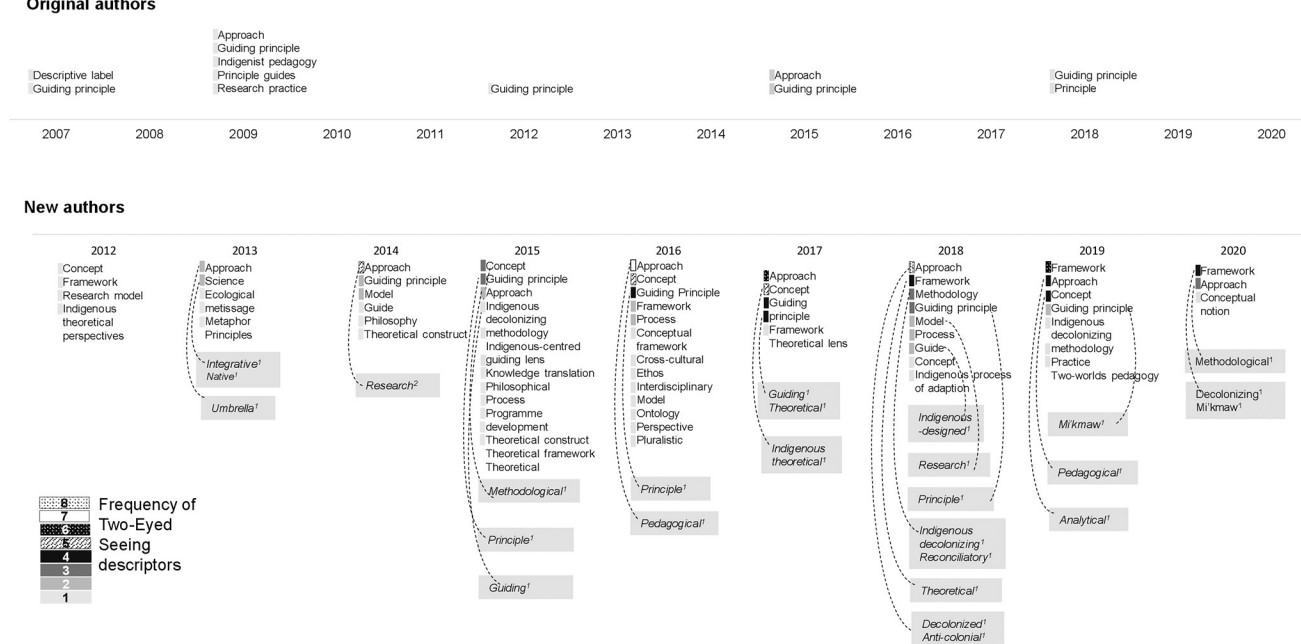

**Fig 2. Two-Eyed Seeing terminologies used by original and new authors over time.** Dotted lines represent qualifiers or extensions of the descriptors that are part of the total number.

were able to group the authors' descriptions into seven key categories. Developing these categories entailed rich conversations amongst team members and with two of the original authors of Two-Eyed Seeing (Elder Albert Marshall and Dr. Cheryl Bartlett), through which we developed a deeper and more nuanced understanding of Two-Eyed Seeing. Below, we outline key crosscutting themes and contradictions that were identified in the review process. Rather than simply describe our findings category-by-category, we have synthesized findings into a critical analysis that encourages readers to think mindfully about the language used to describe Two-Eyed Seeing. The discussion section is broken into three parts: (1) inconsistencies between original and new authors; (2) important observations across categories; and (3) our reflections. In identifying inconsistencies, we are not arguing that certain descriptions are 'right' or 'wrong', nor are we trying to offer the most 'complete' understanding of Two-Eyed Seeing. Rather, we intend to bring different perspectives together to advance our collective understanding. We also want to emphasize that our intention is not to offer a prescriptive formula for how best to apply and use Two-Eyed Seeing. In our conversations with Elder Marshall and Dr. Bartlett, it has become unequivocally clear that there is no "one size fits all" approach and that part of the work of employing the guiding principle is that it compels us to thoughtfully consider each of the characteristics of Two-Eyed Seeing throughout all stages and phases of unique research partnerships, with particular attention to relationship-building, agreed upon understandings of words and language and communication, and how the various parties wish to be involved, represented, and respected throughout the research. This requires an ongoing dialogue that cannot be distilled in a formulaic manner.

## Part 1: Inconsistencies between original and new authors

We noticed that the original authors emphasized some categories—namely Category 5: Spirit and Category 7: Humans being part of ecosystems—more frequently and in greater detail than the new authors. The original authors write that "spirit is at the heart of Indigenous knowledge" [34] and that Two-Eyed Seeing "teaches you to awaken the spirit within you" [104]. According to Elder Albert Marshall, Two-Eyed Seeing encourages individuals to "become a student of life, observant of the natural world. Two-Eyed Seeing teaches that everything is physical and spiritual" [104]. Despite the emphasis on spirit and the natural world, very few new authors discussed the importance of these elements. Most who did, quoted the original authors directly [64, 71]. Bartlett et al. also made reference to this tension: "given that spirit is at the heart of Indigenous knowledge, it would be highly inappropriate if not impossible to ask that mainstream science and much of modern academia—which have diligently scrubbed spirit out of their overall ontology—somehow reverse this diligence" [34].

Similarly, for Category 7: Humans being part of ecosystems, the original authors underscored Two-Eyed Seeing's relevance to the natural environment [28] and the importance of human and non-human knowledge systems to Two-Eyed Seeing [5, 28, 34]. However, this category is not as commonly discussed by the new authors. Two-Eyed Seeing came into the academic world from the disciplines of biology, environmental sciences, and ecology. The unfamiliarity with these subjects and the general lack of education in spiritual knowledge for health researchers and academics might partially explain why it is hard for authors in the health world to make the jump to discussing spiritual and natural components to life. While few new authors write about Two-Eyed Seeing's connection to the natural world, those who have done so acknowledge deep connections between ecology and the environment with the importance of diverse knowledges and ways of knowing. For instance, some new authors write that Two-Eyed Seeing draws attention to relational aspects of complicated issues [37, 64],

reinforces the interconnectedness of both worldviews [43, 63], and supports a process of working together just as all parts of ecosystems must work together [62].

Another notable inconsistency between the original and new authors is that only the new authors describe Two-Eyed Seeing as playing a role in supporting decolonization or Indigenous self-determination (Category 6). Moreover, when the new authors describe Two-Eyed Seeing's place in decolonizing efforts, they do so in very different ways. In some cases, Two-Eyed Seeing supports decolonization by aligning itself to Indigenous research methodologies [80], in others, Two-Eyed Seeing supports the regeneration of Indigenous culture, land and language [71], and in others, Two-Eyed Seeing encourages the bringing in of Indigenous worldviews [82]. Hall et al. similarly consider whether Two-Eyed Seeing supports decolonizing efforts. They write, "The more I read about Two-Eyed Seeing, the more it sounded like decolonizing methodology, but with a different balancing between Western scientific and Indigenous scientific approaches" [58]. Though it is not clear in this article what Hall et al. mean by 'different balancing', it is possible that they are referring to decolonizing methodologies as *privileging* Indigenous ways of knowing and Two-Eyed Seeing as *balancing* diverse ways of knowing. Ultimately, Hall et al. write that Two-Eyed Seeing "aligns with decolonizing and Indigenous research methodologies, governance, and self-determination" because it aligns with the principles of ownership, control, access, and possession (OCAP) and research is ceremony, and links "to Western research traditions in community-based participatory, and participatory research, which have likewise emerged in response to colonial research practices" [58]. The language of 'likewise emerged' reveals a particular point of contention with Two-Eyed Seeing. That is, when Two-Eyed Seeing is described as a decolonizing methodology, it is sometimes misunderstood as emerging as a result of colonialism. It is critical to keep in mind that Two-Eyed Seeing existed prior to colonization; however, it has been given an English name and has been discussed in relation to Western ways of knowing and doing because of our current colonial and neocolonial contexts.

With respect to self-determination, four articles discuss Two-Eyed Seeing's connection with self-determination. Clark writes that Two-Eyed Seeing can contribute to Inuit self-determination as it pertains to electronic health information systems [50, 51], though Clark does not go into detail about what this may look like. Similarly, two articles suggest that Two-Eyed Seeing could contribute to greater Indigenous governance in research processes and data ownership. Hall et al. [58] and Marsh et al. [74] both assert that Two-Eyed Seeing aligns with Indigenous self-determination, particularly with respect to the principles of ownership, control, access, and possession. These articles demonstrate new directions that Two-Eyed Seeing seems to be moving towards—aligning with Indigenous self-determination in some cases, and decolonization in others.

## Part 2: Important observations across categories

**Extensive use of direct quotes.**  We observed that a few classic descriptions of Two-Eyed Seeing were widely cited. For instance, many of the new authors [45, 53, 75] mention the following quote or a variation of it: Two-Eyed Seeing is "learning to see from one eye with the strengths of Indigenous knowledges and ways of knowing, and from the other eye with the strengths of Western knowledges and ways of knowing. . . and learning to use both these eyes together, for the benefit of all" [34]. Other commonly cited passages include: "Two-Eyed Seeing. . .is about life: what you do, what kind of responsibilities you have, how you should live while on Earth" [4] and "Two-Eyed Seeing adamantly, respectfully, and passionately asks that we bring together our different ways of knowing to motivate people, Aboriginal and non-Aboriginal alike, to use all our understandings so that we can leave the world a better place

and not compromise the opportunities for our youth (in the sense of Seven Generations) through our own inaction" [34].

It can be argued that direct quoting has the benefit of staying true to the original literature, which is especially useful for authors who are new to Indigenous health research that uses Two-Eyed Seeing. Nevertheless, Two-Eyed Seeing is a "living knowledge". We understand this to mean that it evolves each time it is characterized and used, that it is not static, and that *knowing* is itself something that is ongoing and ever-changing. As compared to many Euro-western conceptualizations of knowledge, Two-Eyed Seeing views knowledge or *knowing* as verb-based and action-oriented, rather than static and descriptive (a noun) as it is often characterized by Euro-western sciences [105]. Martin [76] notes that as time passes and environments change, so do our perspectives, beliefs, and actions. Therefore, although the use of direct quotes has its merits, it should not be used by authors in place of actively interpreting and critiquing the original literature. When authors describe Two-Eyed Seeing in their own words, new perspectives are generated that demonstrate the author's unique understandings, which, in turn, can be built upon by the readers. We believe that it is only when such healthy cycles of interpretation and critique are established that we can expand upon our shared understanding of Two-Eyed Seeing.

**Relationship between Indigenous and Western knowledges.** The original authors explicitly state that when guided by Two-Eyed Seeing, Indigenous knowledges should not be *tweaked* or *merged* into other knowledges [34]. For instance, Iwama et al. write that "Two-Eyed Seeing neither merges two knowledge systems into one nor does it paste bits of Indigenous knowledge onto Western" [2]. Nevertheless, some new authors have described Two-Eyed Seeing as a *blending* of Indigenous and Western ways of knowing and doing. For instance, Marsh et al. assert that "the blending of Aboriginal and Western research methods, knowledge translation, and program development has been called Two-Eyed Seeing" [17]. Similarly, Whitty-Rogers describes Two-Eyed Seeing as "an umbrella approach that blends our knowledges together—Western and Indigenous" and as "a metaphor for blending scientific and medical knowledge that takes into account the contributions of Aboriginal cultures, science, and experience" [8]. Interestingly, there are sometimes inconsistent renderings of Two-Eyed Seeing even within the same article. For instance, Whitty-Rogers asserts that "blending Western and Indigenous worlds through the Two-Eyed Seeing approach. . .would be a helpful way to understand the Mi'kmaq language, culture, and traditions and restore relationships with each" [8], yet she also writes that "this approach is not meant to blend two cultures, that is the Aboriginal and non-Aboriginal culture, but rather bring together each other's different ways of knowing to try to understand one another" [8]. Whitty-Rogers' statement suggests an awareness that Two-Eyed Seeing is not meant to combine Indigenous and Western cultures into one; nevertheless, the term *blending* suggests just that.

Words such as *blending*, *integrating*, and *merging* imply a mixing of knowledges, which seems contradictory to the original authors' contention that Two-Eyed Seeing should not prompt a merging of knowledges. Other terms like *dance* or a *weaving of knowledges* suggest that there is greater space and opportunity to move between Indigenous and Western ways of knowing and doing. In our interpretation, the latter terms are more aligned with how the original authors discuss Two-Eyed Seeing. The divergent use of *blending* among authors is an example of the power of language, which bears important implications for how Two-Eyed Seeing is understood, used, and even critiqued in Indigenous health research. This is evidenced by another key theme that stood out as we conducted our scoping review—that is, the important role that Two-Eyed Seeing plays in creating space for equitable power relationships between diverse knowledge systems.

**Creating equitable power relationships between Indigenous and Western knowledges.**
In the original authors' articles, Bartlett et al. emphasize that Two-Eyed Seeing recognizes Indigenous and Western knowledges as distinct and whole knowledge systems, which exist side-by-side without the domination of one type of knowledge over another [5]. They write: "Two-Eyed Seeing intentionally seeks to avoid portraying the situation as either 'a clash of knowledges' or 'knowledge domination and assimilation'" [4]. While Two-Eyed Seeing encourages equitable power relationships between Indigenous and Western knowledge systems, it does not prescribe how individuals should navigate between these two ways of knowing nor does it suggest a 50/50 balance between Indigenous and Western knowledges.

Many new authors also note the important role that Two-Eyed Seeing plays in creating space for both perspectives so that one knowledge system does not overpower another [76]. For instance, Hall et al. write that the application of Two-Eyed Seeing acknowledges the inherent power imbalance between Western and Indigenous knowledges that more often than not favour Western knowledge systems. They note that Two-Eyed Seeing is intended to "bridge the divide of power and understanding between Indigenous and Western researchers and processes" [58]. Similarly, Rowan et al. assert that Two-Eyed Seeing "allows the worldviews to remain autonomous, free from knowledge domination and assimilation" [81], and Heath-Engel writes that Two-Eyed Seeing "puts forth the idea that, although there exists a vast array of knowledges concerning how to understand the world, it is important that no single knowledge dominates others" [62]. Since Two-Eyed Seeing is intended to create space for equitable power relationships, it is important that the terms we use, such as *blending*, *bridging*, *or dancing*, etc., reflect this intention so that Indigenous knowledges are not at risk of being dominated or overpowered.

**Diverse terminologies.**   As noted in the Results section, authors used a variety of terms to refer to Two-Eyed Seeing. When we delved deeper into the new authors' descriptions of Two-Eyed Seeing in our qualitative coding, we noticed that the new authors often refer to Two-Eyed Seeing using language commonly found in academic spaces. Some of these terms included: *research model*, *theoretical framework*, and *methodology* [17, 45, 50]. There are good reasons for authors to use terms from their respective disciplines; academic language can help authors both better understand Two-Eyed Seeing and demonstrate their understanding to others. Nevertheless, though these terms can be useful, it is important to keep in mind that Two-Eyed Seeing extends far beyond the technical and methodological aspects of research. For that reason, academic terminologies may, at times, restrict understandings of Two-Eyed Seeing. We want to highlight the tendency to describe Two-Eyed Seeing using various academic terms to suggest that language is important and has implications for how we understand the guiding principle. For instance, referring to Two-Eyed Seeing as a 'theoretical framework' suggests that the guiding principle exists at the level of ontology or epistemology, whereas referring to Two-Eyed Seeing as a 'methodology' or 'process' suggests that the guiding principle exists at the level of methodology. It is important for new authors to be mindful about the terms they use when referring to Two-Eyed Seeing to accurately convey its position (e.g., at an ontological or methodological level) in their understandings.

Indeed, one of the most poignant examples of this is in relation to Martin's (2012) paper, in which she describes Two-Eyed Seeing as a 'mechanism'. This paper was one of the earliest pieces that describes Two-Eyed Seeing that does not include one of the original authors. Since then, Martin (one of the co-authors of our scoping review) has listened to Elder Marshall and Cheryl Bartlett speak on the topic on multiple occasions and has learned that Two-Eyed Seeing is far more than a mechanism (which implies that it is at the level of a tool in one's toolbox that can be drawn upon as needed to solve a problem); instead, it can perhaps be thought of at the level of an ontology—a way of thinking that entirely shapes how one views the world and our

relationship to it. Perhaps one of the most important ways that we can demonstrate that we are continuing to learn is to articulate when our perspectives have deepened or changed in response to new knowledge.

## Part 3: Our reflections and conclusion

A key feature of Two-Eyed Seeing described by the original authors (and many others) is the importance of co-learning. According to the original authors, co-learning implies that the learning that occurs when Indigenous and Western research teams work together must be bi-directional; both parties have the opportunity to learn from one another, thus it is very important for the channels of communication to remain open and transparent and for everyone to agree to work in such a way that everyone's perspectives and views are able to be shared. The imagery that the original authors often use to depict this equitable space for communicating is two people kneeling by a sacred fire. When sitting around a fire everyone is literally on the same level—it creates the space where open communication and transparency can materialize.

Guided by this spirit of co-learning, our engagement with Two-Eyed Seeing literature in Indigenous health research prompted us to engage in deep and rich discussions throughout the scoping review process and to learn together. One key learning arose around the wholeness or partiality of Indigenous and Western knowledges and knowledge systems. For instance, although Bartlett et al. (2012) write that Two-Eyed Seeing recognizes Indigenous and Western knowledges as distinct and whole knowledge systems, the authors also suggest that in bringing together Indigenous and Western knowledges, greater understandings of a topic area may result, suggesting that each understanding of the world is partial. Initially, these descriptions seemed contradictory to our team. Some team members asked: If Indigenous and Western knowledges are whole of themselves, do they *need* Western and/or Indigenous knowledges to 'complement' them? After bringing two knowledge systems together, is the new knowledge more *complete*?

New authors who describe Two-Eyed Seeing also try to distinguish between the wholeness vs. partiality of knowledge. Latimer et al. write that Two-Eyed Seeing "recognizes the overlap between two distinct yet evolving knowledge systems" [68] and, when brought together, these distinct knowledge systems provide a richer understanding of the world. Similarly, Martin asserts that Two-Eyed Seeing understands the Indigenous 'eye' and Western 'eye' as representing partial ways of seeing the world. She writes: "When both eyes are used together, this does not mean that our view is now "complete and whole," but a new way of seeing the world has been created—one that respects the differences that each can offer" [76].

Upon greater discussion, our team determined that there is indeed no contradiction. We understand Two-Eyed Seeing as recognizing Indigenous and Western *knowledge systems* as whole and distinct in and of themselves, and at the same time, Two-Eyed Seeing holds that each knowledge system can only offer a partial understanding of the world. That is, Two-Eyed Seeing operates under the assumption that even though Indigenous and Western knowledge systems are whole in and of themselves, no single worldview offers everything. For that reason, individuals can benefit from the bringing together of different *knowledges* and *ways of knowing* in order to arrive at a richer—though still partial—understanding of the world.

## Limitations of our study

It is important to note that Elders Albert and Murdena Marshall have not published a great deal in peer review journals. Many of the mediums they use to discuss Two-Eyed Seeing are visual or oral, such as videos, diagrams, lectures, or in-person discussions, and in some of these mediums, the Elders describe Two-Eyed Seeing in their Mi'kmaq language (see S2 File

for list of presentations). Toward the end of our scoping review, we invited Elder Albert and Dr. Cheryl Bartlett to review this manuscript. Their comments confirmed that a major limitation to our study is the sole focus on English-language written texts, since we did not have the resources to analyze visual outputs or texts in other languages. Future studies may want to investigate descriptions of Two-Eyed Seeing using non-written mediums and in the Mi'kmaq language.

Another limitation to our study is that we used the JBI scoping review methodology, which does not explicitly include Indigenous knowledges or ways of knowing. It is possible that using a methodology that comes from a Western European worldview limited us from 'seeing' things beyond what was outlined in the JBI Manual. What's more, even though we sought to go beyond the JBI methodology and to capture aspects of Two-Eyed Seeing through a thematic analysis, we approached the analysis from a Western perspective. Though we tried to pick up on nuanced descriptions of Two-Eyed Seeing in our thematic analysis and not to homogenize characterizations of the guiding principle [106], we grouped Two-Eyed Seeing into categories, which required us to overlook 'small' differences. Future research may wish to examine descriptions of Two-Eyed Seeing using a process explicitly grounded in Indigenous knowledges and worldviews.

## Conclusion

Our review extends upon recent integrative and scoping reviews [7, 14] by exploring the meaning of Two-Eyed Seeing in seven descriptive categories, and comparing the language that both the original and new authors use. The heterogenous results on terminology use and different emphasis on certain categories (namely spirit, humans being part of ecosystems, and decolonizing and self-determination) suggest that Two-Eyed Seeing might have been interpreted in ways that the original authors had not initially envisioned. This review is timely as more researchers become interested in Two-Eyed Seeing and are looking to deepen their understanding of this guiding principle.

## Supporting information

**S1 File. Etuaptmumk / Two-Eyed Seeing for knowledge gardening.** Original manuscript by Elder Albert Marshall and Dr. Cheryl Bartlett in *Encyclopedia of Educational Philosophy and Theory* (Springer online) edited by Michael A. Peters within section "Indigenous Education in Canada" coordinated by Michelle Hogue.
(PDF)

**S2 File. Recommended readings.** List of Elder Albert Marshall and Elder Murdena Marshall's presentations, panels, and publications about Two-Eyed Seeing.
(DOC)

**S1 Table. Search strategies.** Sources of information and number of results retrieved.
(DOCX)

**S2 Table. Characteristics of articles describing Two-Eyed Seeing by original authors.** Year of publication, study participants, geographical location, research design and aim of study.
(DOCX)

**S3 Table. Characteristics of articles describing Two-Eyed Seeing by new authors.** Year of publication, study participants, geographical location, research design and aim of study.
(DOCX)

**S4 Table. Thematic analysis of original authors' descriptions of Two-Eyed Seeing.** Mapping extracted quotations to descriptive categories.
(DOCX)

**S5 Table. Thematic analysis of new authors' descriptions of Two-Eyed Seeing.** Mapping extracted quotations to descriptive categories.
(DOCX)

## Acknowledgments

We would like to thank Elder Albert Marshall and Dr. Cheryl Bartlett for their generosity, patience, kindness, and commitment to sharing and teaching about their work about Two-Eyed Seeing with us. Wela'lieg for taking time to review and provide comments to this manuscript and for continually broadening and enriching our understandings of the guiding principle. Wela'lieg to Elder Albert for sharing Mi'kmaw knowledge to pass on from generation to generation.

## Author Contributions

**Conceptualization:** Anita C. Benoit.

**Data curation:** Sophie I. G. Roher, Ziwa Yu.

**Formal analysis:** Sophie I. G. Roher, Ziwa Yu, Debbie H. Martin, Anita C. Benoit.

**Funding acquisition:** Debbie H. Martin.

**Methodology:** Ziwa Yu.

**Project administration:** Sophie I. G. Roher.

**Software:** Ziwa Yu.

**Supervision:** Debbie H. Martin, Anita C. Benoit.

**Validation:** Debbie H. Martin.

**Visualization:** Anita C. Benoit.

**Writing – original draft:** Sophie I. G. Roher, Ziwa Yu, Debbie H. Martin, Anita C. Benoit.

**Writing – review & editing:** Sophie I. G. Roher, Ziwa Yu, Debbie H. Martin, Anita C. Benoit.

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
