## [Decision Letter · Decision Letter 0]

1 Mar 2021

PONE-D-20-35395

How is Etuaptmumk/Two-Eyed Seeing Being Used in Indigenous Health Research? A Scoping Review.

PLOS ONE

Dear Ms. Roher,

Thank you for submitting your manuscript to PLOS ONE. After careful consideration, we feel that it has merit but does not fully meet PLOS ONE’s publication criteria as it currently stands. Therefore, we invite you to submit a revised version of the manuscript that addresses the points raised during the review process.

Please see the Overarching Recommendations for revision below as well as the two reviews below.  Please address all comments.

We look forward to receiving your revised manuscript.

Kind regards,

Sarah E.P. Munce, PhD

Academic Editor

PLOS ONE

Journal Requirements:

2.In your Data Availability statement, you have not specified where the minimal data set underlying the results described in your manuscript can be found. PLOS defines a study's minimal data set as the underlying data used to reach the conclusions drawn in the manuscript and any additional data required to replicate the reported study findings in their entirety. All PLOS journals require that the minimal data set be made fully available. For more information about our data policy, please see http://journals.plos.org/plosone/s/data-availability.

Additional Editor Comments:

Academic Editor Notes:

The other reviewer has conducted an excellent review and I agree that the Discussion could be a bit more succinct and include some very practical suggestions based on the findings of the review for how Two-Eyed Seeing can be adopted in health research (see comments below). The supplementary files should also be pared down please.

This is a well-written, interesting, and comprehensive review. The authors are to be commended for their great efforts! My own review of this excellent review is included below:

Introduction

It is recommended that the Introduction be reorganized in a few areas (detailed below). One area includes the statements on the purposes of the review: they should be placed right before the Methods rather than on pages 4-5. There is no need to repeat the purpose statements.

When the authors state on page 6, “Each of these terms carries very different meanings and has consequences for how the guiding principle is used in research studies”, the authors should delineate specifically how these terms have different meanings and the associated consequences (even with 1-2 sentences of the use of i.e., )

For the sentence, “When new concepts or frameworks are introduced into academia and taken on in various research projects, it is not unusual that they would morph and transform with time and context” (page 6), it would be helpful if the authors included some references here and some brief examples.

The description of the reviews that have been previously conducted should be included before the rationale for the current review; this reorganization would further strengthen the rationale for the current review (i.e., this is what has been done previously and this is how the current review is different and will contribute to this area of research). The rationale for the current review is otherwise very clear.

Methods

The statement, “We also used the Preferred Reporting Items for Systematic Reviews and Meta-analyses extension for Scoping Reviews (PRISMA-ScR) (27)” is incomplete. It should read, if true, “We also used the PRISMA-ScR to guide the conduct and reporting of this scoping review”. It should also be included in the first paragraph in the Methods section.

The authors indicate in their Extraction section that they extracted on “…whether the term Etuaptmumk was used”. Could the authors clarify this – as I assumed that using the term Etuaptmumk was used was part of the inclusion criteria for the review itself (i.e., all included articles would have used the term).

More details on the process of “qualitative thematic coding” are needed (e.g., applicable components of the Data Analysis section of the COREQ checklist). For example, how many coders were there? Was a software such as NVivo used to organize the data?

Results

The authors indicate that “Most research included in this review was based in Canada (n = 71). Among these, the majority were in Ontario (n = 16) and Nova Scotia 278 (n = 14)”. Sixteen studies from Ontario out of 71 does not seem like the majority. Could the authors clarify this please?

Discussion

The Discussion section is very thoughtful.

Based on the findings of the review, could the authors add some very practical suggestions/recommendations for health researchers who adopt Two-Eyed Seeing as a guiding principle e.g., how it can be applied/adopted? Specifically outline the (potential) benefits to using this approach, etc.? Minimum criteria to indicate that this approach has been adopted? This could be especially helpful when reporting on the use of Two-Eyed Seeing and future authors who wish to replicate and advance approaches.

Minor Points

The use of the term “unpack” reads as a little colloquial – I suggest changing it to “investigate” or “explore” or “determine”.

The statement, “This is particularly evident for research approaches that do not adhere to a checklist, …” (page 6) could be modified to “This is often observed when associated guidelines for their use have not been developed or implemented” (i.e., adhering to a checklist reads as very specific).

The use of the term “watered down” reads as colloquial and could be replaced with “diminished”.

It is suggested that the statement “Our scoping review seeks to do just that” (page 7) should be removed.

On page 18, the sentence “It is one of the first article to discuss ‘health research’ directly…” should read “It is one of the first articles to discuss…”

Please see above regarding use of “unpack”. Similarly, the phrase “struck us” also reads as colloquial and should be replaced with more formal writing please.

Page 22: “…mitigate against…”; the against is not needed.

Page 24: The authors should remove “admittedly” and could replace it with “It is argued that…”

Page 24: It is not clear what the authors mean by “living knowledge” – please clarify.

Reviewers' comments:

Reviewer's Responses to Questions

**Comments to the Author**

1. Is the manuscript technically sound, and do the data support the conclusions?

Reviewer #1: Yes

2. Has the statistical analysis been performed appropriately and rigorously? 

Reviewer #1: Yes

3. Have the authors made all data underlying the findings in their manuscript fully available?

Reviewer #1: Yes

4. Is the manuscript presented in an intelligible fashion and written in standard English?

Reviewer #1: Yes

5. Review Comments to the Author

Reviewer #1: The authors conducted text analyses of numerous articles to distill central meaning of Etauptmumk, translated as Two-Eyed Seeing, and to consider their relevance for Indigenous health research. The work is aligned with the Canadian Institutes of Health Research (CIHR) strategic plan for Aboriginal People’s Health. Overall, the Etauptmumk framework has wide relevance for health research in general, given ways in which it complements and sometimes challenges Western approaches to health.

The populations of interest in CIHR Indigenous health research include First Nations, Inuit, or Métis communities and their cultures, experiences, and knowledge systems, past and present. Using state of the art databases and screening criteria, the authors identified a large number of articles, including diverse types of studies and reviews across a variety of academic disciplines (health sciences, education, biology, etc.). Clearly formulated eligibility criteria resulted in selection of 80 articles, from which quotations were extracted and subjected to extensive text analysis to distill key meaning of Etauptmumk.

Seven categories of meaning were identified, each of which is richly described. A major message in the first meaning, guide for life, is the broad purview of Two-Eyed Seeing: it is a wholistic way of knowing, being, doing and seeing that is mental, spiritual, physical, and emotional; it does not fit into a particular subject area or discipline, but rather “is about life – what you do, what kind of responsibilities you have, how you should live while on Earth – as a guiding principle that covers all aspects of our lives: social, economic, environmental” (p.13). Importantly, these ideas are largely missing in Western approaches to health, which exist mostly in disciplinary silos that do not engage with each other. The meaning of Western health is thus fundamentally fragmented in comparison to Etauptmumk.

The meaning, responsibility for the greater good and future generations, calls for using all capacities (gifts) and actions to leave the world a better place. This overarching value is also rarely embraced in non-Indigenous health research. The meaning, co-learning process, emphasizes the importance of relationship-building by having different peoples put their own knowledge and actions forward for examination, with recognition that there might not be consensus. A related meaning, multiple or diverse perspectives, underscores respect for and acceptance of diverse realities. These ideas are important for recognizing Indigenous knowledge as valid and useful in its own right: “Two-eyed seeing does not subsume one way of knowing over another” (p.16). Such awareness allows for an interweaving (described as a “dance”) between different forms of knowledge. Spirit is a central meaning that is seen as universal (there is a spirit in everything) and essential for a complete person involving interaction of body, mind, soul and spirit with all aspects of nature. Key for honoring Indigenous perspectives is decolonization and self-determination, a meaning that underscores principles of ownership control, access, and possession. Lastly, the meaning, humans as part of ecosystems, sees human health as requiring balance and integrity between people and the global ecosystems that surround them.

Detailed descriptions of the above meanings are followed with a thoughtful Discussion that highlights differences between in meanings between original publications on Etauptmumk and more recent studies. For example, seeing human beings seen as part of ecosystems is more frequently detailed in the work of new authors, whereas emphasis about spiritual knowledge for health is more prominent in earlier writings about Two-Eyed Seeing. Importantly, the relationships between Indigenous and Western knowledge of health are also covered, although this topic warrants greater emphasis, given that Western health research is notably deficient in most of the seven meanings. That is to say, Indigenous perspectives on health are not relevant exclusively for First Nations peoples because they reveal important counterpoints and omissions in health research among non-Indigenous peoples. Other topics in the Discussion (e.g., Diverse Terminologies) seem unnecessary, given attention to such topics in the introductory section. In general, the lengthy Discussion (13 pages) would benefit from being distilled to a more succinct summary of what the comprehensive text analyses revealed and the relevance of the meanings of Two-Eyed Seeing for understanding the health and well-being of Indigenous peoples as well as for highlighting omissions in culturally dominant Western approaches to health, which are increasingly concerned with molecular science, big data and advances in machine-learning. The juxtaposition of these differing approaches is worthy of reflection by all health researchers.

A related suggestion is to pare down the supportive materials included with the manuscript – nine large files are currently included, including details lists of excluded articles, search strategies, and lengthy tables and appendices of the articles included. Much of this material might be better noted as available for review from the authors so as to keep the focus on targeted tables that are directly pertinent to the analyses conducted.

6. PLOS authors have the option to publish the peer review history of their article (what does this mean?). If published, this will include your full peer review and any attached files.

Reviewer #1: No

---

## [Author Response · Author response to Decision Letter 0]

2 May 2021

Dear Dr. Sarah Munce, 

Thank you for considering our manuscript “How is Etuaptmumk/Two-Eyed Seeing Characterized in Indigenous Health Research? A Scoping Review” for publication and for sending along the reviewers’ very helpful comments. We feel we have addressed the feedback in the attached revised manuscript. These changes are explained in detail below.

Please note that we have included the following items when submitting our revised manuscript:

• This rebuttal letter, which responds to each point raised by the academic editor and reviewers. We have uploaded this letter as a separate file labeled 'Response to Reviewers'.

• A marked-up copy of the manuscript that highlights changes made to the original version. We uploaded this as a separate file labeled 'Revised Manuscript with Track Changes'.

• An unmarked version of the revised paper without tracked changes. We uploaded this as a separate file labeled 'Manuscript'.

• The page numbers in this cover letter reflect those in the marked version of the revised paper with track changes. 

• We made changes to our financial disclosure and included the updated statement in our cover letter. 

We have provided an itemized list of our responses to the reviewers’ comments below: 

1. Journal Requirements:

a) Please review your reference list to ensure that it is complete and correct. 

Response: We have reviewed our reference list to ensure that it is complete and correct. We noticed some inconsistencies (such as citing the same paper twice) and have made changes to correct these errors. 

b) If you have cited papers that have been retracted, please include the rationale for doing so in the manuscript text, or remove these references and replace them with relevant current references. 

If you need to cite a retracted article, indicate the article’s retracted status in the References list and also include a citation and full reference for the retraction notice. 

Response: We have not included any cited papers that have been retracted. 

The PLOS ONE style templates can be found at

Response: We have ensured the manuscript meets PLOS ONE’s style requirements, including those for file naming.

b) In your Data Availability statement, you have not specified where the minimal data set underlying the results described in your manuscript can be found. PLOS defines a study's minimal data set as the underlying data used to reach the conclusions drawn in the manuscript and any additional data required to replicate the reported study findings in their entirety. All PLOS journals require that the minimal data set be made fully available. For more information about our data policy, please see http://journals.plos.org/plosone/s/data-availability. 

Response: We have changed our data availability statement to specify: “All relevant data are within the manuscript and its Supporting Information files.”

Response: The minimal underlying data set is uploaded in the supporting information files S2-6. As mentioned above, we have changed our data availability statement to specify: “All relevant data are within the manuscript and its Supporting Information files.”

Response: there are no ethical or legal restrictions to sharing our data publicly. 

Response: We have changed our data availability statement to specify: “All relevant data are within the manuscript and its Supporting Information files.”

f) Please include captions for your Supporting Information files at the end of your manuscript, and update any in-text citations to match accordingly. Please see our Supporting Information guidelines for more information: http://journals.plos.org/plosone/s/supporting-information. 

Response: List of supporting information files and captions have been inserted at the end of manuscript.

3. Academic Editor Notes

a) The Discussion could be a bit more succinct 

Response: We have shortened the discussion. The specific changes are addressed below (see our response to 4c).

b) Include some very practical suggestions based on the findings of the review for how Two-Eyed Seeing can be adopted in health research (see comments below). 

Response: Thank you for this comment. It has helped us to clarify our intentions for this paper. The purpose of this paper is to describe how new and original authors have defined Two-Eyed seeing. In this paper, we are not intending to identify how Two-Eyed Seeing is being applied and adopted in health research. We see this as a separate issue that we are exploring in a separate paper. For that reason, we have changed the title of the paper to “How is Etuaptmumk/Two-Eyed Seeing Characterized in Indigenous Health Research? A Scoping Review.” (The former title was “How is Etuaptmumk/Two-Eyed Seeing Being Used in Indigenous Health Research? A Scoping Review”). To clarify our intentions in this paper, we have also removed language that discusses ‘applications’ of Two-Eyed Seeing in the introduction. We have also added some statements throughout the manuscript to clarify our intentions. For instance, on page 5 of the introduction we added the sentence, “Rather than identify how Two-Eyed Seeing is being applied and adopted in health research, our study seeks to describe and compare the language used by the original and new authors to represent Two-Eyed Seeing.” 

c) The supplementary files should also be pared down please.

Response: Thank you for this comment. We have made the changes described below to the supplementary material. Instead of 11 supplementary documents we now have 7 documents. Although there are no restrictions on tables and word count, we appreciate the opportunity to be more concise with our tables and figures which has improved the quality of our manuscript.

- Supplementary Tables 2 (S2, protocol) and 4 (S4, specific articles and reasons) have been removed as the information is currently well-described in the manuscript. 

- Supplementary 5 (S5) Figure 1 will be removed from the supplementary material and included as Figure 1 in the body of the manuscript if accepted for publication. Supplementary 10 (S10) Figure 2 will also be included as part of the body of the manuscript.

- Four supplementary documents (S3-S6) are critical tables of our findings which we cannot remove. We have identified 80 eligible articles which will impact on the size and number of our tables. 

- S1 and S7 are information regarding two-eyed seeing that if removed would not be available outside this study. S1 specifically is the unedited version of a definition that is preferred by its authors.

d) Introduction: It is recommended that the Introduction be reorganized in a few areas (detailed below). One area includes the statements on the purposes of the review: they should be placed right before the Methods rather than on pages 4-5. There is no need to repeat the purpose statements. 

Response: Thank you for this comment. We have re-organized the introduction. The purposes of the review have been placed before the methods and the repeated purpose statements were deleted. 

e) Introduction: When the authors state on page 6, “Each of these terms carries very different meanings and has consequences for how the guiding principle is used in research studies”, the authors should delineate specifically how these terms have different meanings and the associated consequences (even with 1-2 sentences of the use of i.e.)

Response: Thank you for this comment. We have added the following sentences to the manuscript on page 5: “For example, a methodology is largely focused on technique, whereas a theoretical framework may provide the logic behind methodological choices. While both provide insight into complex and competing worldviews, characterizing Two-Eyed Seeing as a methodology versus a theoretical framework can imply very different usage based on their definitions.”

f) Introduction: For the sentence, “When new concepts or frameworks are introduced into academia and taken on in various research projects, it is not unusual that they would morph and transform with time and context” (page 6), it would be helpful if the authors included some references here and some brief examples.

Response: We appreciate this comment as providing examples will strengthen our manuscript. We have provided two examples to address this comment. See below for the additional text added to the manuscript:

Page 6: “For example, intersectionality is a concept emerging from ideas debated in critical race theory (21). It has since then been adapted to meet diverse disciplinary needs, leading many scholars to engage with it theoretically and methodologically (22). Similarly, feminist ideologies and movements have evolved to include several waves recognizing white feminist theories and later multiracial feminist theories.”

g) Introduction: The description of the reviews that have been previously conducted should be included before the rationale for the current review; this reorganization would further strengthen the rationale for the current review (i.e., this is what has been done previously and this is how the current review is different and will contribute to this area of research). The rationale for the current review is otherwise very clear. 

Response: Thank you for this comment. We have reorganized the introduction section so that the reviews that have been previously conducted are included before we discuss the rationale for the current review. 

h) Methods: The statement, “We also used the Preferred Reporting Items for Systematic Reviews and Meta-analyses extension for Scoping Reviews (PRISMA-ScR) (27)” is incomplete. It should read, if true, “We also used the PRISMA-ScR to guide the conduct and reporting of this scoping review”. It should also be included in the first paragraph in the Methods section. 

Response: Thank you. We have moved the PRISMA-ScR statement to the first paragraph under Methods.

i) Methods: The authors indicate in their Extraction section that they extracted on “…whether the term Etuaptmumk was used”. Could the authors clarify this – as I assumed that using the term Etuaptmumk was used was part of the inclusion criteria for the review itself (i.e., all included articles would have used the term).

Response: Thank you for this comment. Etuaptmumk was a term used as part of the search strategy with the Boolean operator ‘OR’, but not strictly part of the inclusion criteria. Therefore, eligible articles could use the term ‘Two-Eyed Seeing’ OR ‘Etuaptmumk’. Through our knowledge of the literature and while developing the search strategy, it was clear that Etauptmumk is not used consistently, is infrequently mentioned in conjunction with Two-Eyed Seeing, and was never used without the term Two-Eyed Seeing. We have not included this explanation in the manuscript. However, if you feel that it should be included within the manuscript, we are happy to make this change. 

j) Methods: More details on the process of “qualitative thematic coding” are needed (e.g., applicable components of the Data Analysis section of the COREQ checklist). For example, how many coders were there? Was a software such as NVivo used to organize the data? 

Response: Thank you for this comment. We outline on page 9 in the revised manuscript that two reviewers independently completed coding and then the entire team met to reach consensus. We used Excel to manage all steps in data extraction.

k) Results: The authors indicate that “Most research included in this review was based in Canada (n = 71). Among these, the majority were in Ontario (n = 16) and Nova Scotia 278 (n = 14)”. Sixteen studies from Ontario out of 71 does not seem like the majority. Could the authors clarify this please? 

Response: We changed the sentence to “Among these, 16 were in Ontario and 14 were in Nova Scotia.” (pg. 12)

l) Discussion: The Discussion section is very thoughtful. Based on the findings of the review, could the authors add some very practical suggestions/recommendations for health researchers who adopt Two-Eyed Seeing as a guiding principle e.g., how it can be applied/adopted? Specifically outline the (potential) benefits to using this approach, etc.? Minimum criteria to indicate that this approach has been adopted? This could be especially helpful when reporting on the use of Two-Eyed Seeing and future authors who wish to replicate and advance approaches. 

Response: We believe that the crux of this comment has been addressed in 3.B) (above). We have attempted to further clarify our intention for this paper, which is not to offer ways in which to apply or adopt Two-Eyed Seeing. Our research team talked at length about the importance of further emphasizing that there is no “one size fits all approach” to encouraging the use or application of Two-Eyed Seeing, since it is entirely context/content/culturally specific. This makes us reluctant to include ways to replicate it. We have included wording to this effect on pages 19 in the discussion.

m) Minor Points: The use of the term “unpack” reads as a little colloquial – I suggest changing it to “investigate” or “explore” or “determine”. 

Response. Thank you for this comment. We made this change. 

n) Minor Points: The statement, “This is particularly evident for research approaches that do not adhere to a checklist, …” (page 6) could be modified to “This is often observed when associated guidelines for their use have not been developed or implemented” (i.e., adhering to a checklist reads as very specific).

Response. Thank you for this comment. We made this change.

o) Minor Points: The use of the term “watered down” reads as colloquial and could be replaced with “diminished”. 

Response. Thank you for this comment. We made this change. 

p) Minor Points: It is suggested that the statement “Our scoping review seeks to do just that” (page 7) should be removed. 

Response. Thank you for this comment. We made this change. 

q) Minor Points: On page 18, the sentence “It is one of the first article to discuss ‘health research’ directly…” should read “It is one of the first articles to discuss…” 

Response. Thank you for this comment. We made this change. 

r) Minor Points: Please see above regarding use of “unpack”. Similarly, the phrase “struck us” also reads as colloquial and should be replaced with more formal writing please. 

Response. Thank you for this comment. We changed the sentence to: “Below, we outline key crosscutting themes and contradictions that came up in the review process.”

s) Minor Points: Page 22: “…mitigate against…”; the against is not needed. 

Response. Thank you for this comment. We made this change. 

t) Minor Points: Page 24: The authors should remove “admittedly” and could replace it with “It is argued that…” 

Response. Thank you for this comment. We changed the sentence with “it can be argued that...”

u) Minor Points: Page 24: It is not clear what the authors mean by “living knowledge” – please clarify. 

Response. Thank you for this comment. We have provided further explanation on page 23-24 what is meant by “living knowledge”.

4. Reviewer #1.

a) The authors conducted text analyses of numerous articles to distill central meaning of Etauptmumk, translated as Two-Eyed Seeing, and to consider their relevance for Indigenous health research. The work is aligned with the Canadian Institutes of Health Research (CIHR) strategic plan for Aboriginal People’s Health. Overall, the Etauptmumk framework has wide relevance for health research in general, given ways in which it complements and sometimes challenges Western approaches to health.

The populations of interest in CIHR Indigenous health research include First Nations, Inuit, or Métis communities and their cultures, experiences, and knowledge systems, past and present. Using state of the art databases and screening criteria, the authors identified a large number of articles, including diverse types of studies and reviews across a variety of academic disciplines (health sciences, education, biology, etc.). Clearly formulated eligibility criteria resulted in selection of 80 articles, from which quotations were extracted and subjected to extensive text analysis to distill key meaning of Etauptmumk.

Seven categories of meaning were identified, each of which is richly described. A major message in the first meaning, guide for life, is the broad purview of Two-Eyed Seeing: it is a wholistic way of knowing, being, doing and seeing that is mental, spiritual, physical, and emotional; it does not fit into a particular subject area or discipline, but rather “is about life – what you do, what kind of responsibilities you have, how you should live while on Earth – as a guiding principle that covers all aspects of our lives: social, economic, environmental” (p.13). Importantly, these ideas are largely missing in Western approaches to health, which exist mostly in disciplinary silos that do not engage with each other. The meaning of Western health is thus fundamentally fragmented in comparison to Etauptmumk.

The meaning, responsibility for the greater good and future generations, calls for using all capacities (gifts) and actions to leave the world a better place. This overarching value is also rarely embraced in non-Indigenous health research. The meaning, co-learning process, emphasizes the importance of relationship-building by having different peoples put their own knowledge and actions forward for examination, with recognition that there might not be consensus. A related meaning, multiple or diverse perspectives, underscores respect for and acceptance of diverse realities. These ideas are important for recognizing Indigenous knowledge as valid and useful in its own right: “Two-eyed seeing does not subsume one way of knowing over another” (p.16). Such awareness allows for an interweaving (described as a “dance”) between different forms of knowledge. Spirit is a central meaning that is seen as universal (there is a spirit in everything) and essential for a complete person involving interaction of body, mind, soul and spirit with all aspects of nature. Key for honoring Indigenous perspectives is decolonization and self-determination, a meaning that underscores principles of ownership control, access, and possession. Lastly, the meaning, humans as part of ecosystems, sees human health as requiring balance and integrity between people and the global ecosystems that surround them.

Detailed descriptions of the above meanings are followed with a thoughtful Discussion that highlights differences between in meanings between original publications on Etauptmumk and more recent studies. For example, seeing human beings seen as part of ecosystems is more frequently detailed in the work of new authors, whereas emphasis about spiritual knowledge for health is more prominent in earlier writings about Two-Eyed Seeing. Importantly, the relationships between Indigenous and Western knowledge of health are also covered, although this topic warrants greater emphasis, given that Western health research is notably deficient in most of the seven meanings. That is to say, Indigenous perspectives on health are not relevant exclusively for First Nations peoples because they reveal important counterpoints and omissions in health research among non-Indigenous peoples. 

Response: Thank you for this wonderful summary of the paper! 

b) Other topics in the Discussion (e.g., Diverse Terminologies) seem unnecessary, given attention to such topics in the introductory section. 

Response: Thank you for this comment. We have removed the diverse terminologies section in the introduction.

c) In general, the lengthy Discussion (13 pages) would benefit from being distilled to a more succinct summary of what the comprehensive text analyses revealed and the relevance of the meanings of Two-Eyed Seeing for understanding the health and well-being of Indigenous peoples as well as for highlighting omissions in culturally dominant Western approaches to health, which are increasingly concerned with molecular science, big data and advances in machine-learning. The juxtaposition of these differing approaches is worthy of reflection by all health researchers. 

Response: We have distilled the discussion into a more succinct summary of the findings (previously it was 13 pages and now it is 11). 

d) A related suggestion is to pare down the supportive materials included with the manuscript – nine large files are currently included, including details lists of excluded articles, search strategies, and lengthy tables and appendices of the articles included. Much of this material might be better noted as available for review from the authors so as to keep the focus on targeted tables that are directly pertinent to the analyses conducted.

Response: Thank you for this comment. Please see our response for 3c.

---

## [Editor Report · Decision Letter 1]

11 Jun 2021

PONE-D-20-35395R1

How is Etuaptmumk/Two-Eyed Seeing Characterized in Indigenous Health Research? A Scoping Review.

PLOS ONE

Dear Ms. Roher,

Thank you again for submitting your manuscript to PLOS ONE. Your paper has some final recommended revisions. Therefore, we invite you to submit a revised version of the manuscript that addresses these points.

We look forward to receiving your revised manuscript.

Kind regards,

Sarah E.P. Munce, PhD

Academic Editor

PLOS ONE

Journal Requirements:

Additional Editor Comments (if provided):

The authors are to commended for systematically addressing the comments. Please see some minor, final recommendations below.

Abstract

In the abstract, please write out the Joanna Briggs Institute rather than use the acronym JBI, which has not been previously introduced.

Conclusion

Please remove the sentence: “Once again, the purpose of highlighting these differences is not to judge or criticize, but to encourage a larger dialogue about Two-Eyed Seeing and the impact of language on how the guiding principle is understood”.

Rather than “This reflection is timely…” please replace with “This review is timely …”

Please remove the sentence: “We hope that this review will assist aspiring researchers in their learning journey”.

---

## [Author Response · Author response to Decision Letter 1]

14 Jun 2021

Dear Dr. Sarah Munce, 

Thank you for considering our manuscript “How is Etuaptmumk/Two-Eyed Seeing Characterized in Indigenous Health Research? A Scoping Review” for publication and for sending along the editor’s comments. We have addressed the editor’s feedback in the attached revised manuscript. The changes are explained below.

Please note that we have included the following items when submitting our revised manuscript:

• This rebuttal letter, which responds to each point raised by the academic editor and reviewers. We have uploaded this letter as a separate file labeled 'Response to Reviewers'.

• A marked-up copy of the manuscript that highlights changes made to the original version. We uploaded this as a separate file labeled 'Revised Manuscript with Track Changes'.

• An unmarked version of the revised paper without tracked changes. We uploaded this as a separate file labeled 'Manuscript'.

We have provided an itemized list of our responses to the editor’s comments below: 

1. Abstract. In the abstract, please write out the Joanna Briggs Institute rather than use the acronym JBI, which has not been previously introduced.

Response: Thank you for your comment. One of our co-authors (Ziwa Yu) is a JBI-certified reviewer. She was notified in Jan 21, 2021 from JBI’s Global Engagement Administration Officer that JBI went through a rebranding process and discontinued the use of Joanna Briggs Institute. They specifically requested that we use only the term ‘JBI’ when referring to the research organization. 

2. Conclusion

Please remove the sentence: “Once again, the purpose of highlighting these differences is not to judge or criticize, but to encourage a larger dialogue about Two-Eyed Seeing and the impact of language on how the guiding principle is understood”.

Response: We removed the sentence. 

3. Rather than “This reflection is timely…” please replace with “This review is timely …”

Response: We made this change. 

4. Please remove the sentence: “We hope that this review will assist aspiring researchers in their learning journey”.

Response: We removed the sentence.

Response: We uploaded the figures to PACE, which confirmed that they were in the correct format.

---

## [Editor Report · Decision Letter 2]

30 Jun 2021

How is Etuaptmumk/Two-Eyed Seeing Characterized in Indigenous Health Research? A Scoping Review.

PONE-D-20-35395R2

Dear Ms. Roher,

We’re pleased to inform you that your manuscript has been judged scientifically suitable for publication and will be formally accepted for publication once it meets all outstanding technical requirements.

Kind regards,

Sarah E.P. Munce, PhD

Academic Editor

PLOS ONE

---

## [Editor Report · Acceptance letter]

7 Jul 2021

PONE-D-20-35395R2 

How is *Etuaptmumk/*Two-Eyed Seeing characterized in Indigenous health research? A scoping revie 

Dear Dr. Roher:

I'm pleased to inform you that your manuscript has been deemed suitable for publication in PLOS ONE. Congratulations! Your manuscript is now with our production department. 

Kind regards, 

on behalf of

Dr. Sarah E.P. Munce 

Academic Editor

PLOS ONE